# Colistin Resistance Mechanism in *Enterobacter hormaechei* subsp. *steigerwaltii* Isolated from Wild Boar (*Sus scrofa*) in France

**DOI:** 10.3390/pathogens11091022

**Published:** 2022-09-07

**Authors:** Younes Laidoudi, Edgarthe Priscilla Ngaiganam, Jean-Lou Marié, Isabelle Pagnier, Jean-Marc Rolain, Seydina Mouhamadou Diene, Bernard Davoust

**Affiliations:** 1Aix Marseille Université, IRD, APHM, MEPHI, IHU-Méditerranée Infection, 13005 Marseille, France; 2IHU-Méditerranée Infection, 13005 Marseille, France; 3Animal Epidemiology Expert Group, French Military Health Service, 37076 Tours, France

**Keywords:** gene inactivation, *phoP/Q*, *pmrA/B*, *mgrB* regulator, colistin resistance, wild boar, France

## Abstract

Wild animals may act as efficient antimicrobial-resistance reservoirs and epidemiological links between humans, livestock, and natural environments. By using phenotypic and genotypic characterization, the present study highlighted the occurrence of an antimicrobial-resistant (i.e., amoxicillin, amoxicillin–clavulanic acid, cephalothin, and colistin) *Enterobacter hormaechei* subsp. *steigerwaltii* strain in wild boar (*Sus scrofa*) from France. The molecular analysis conducted showed non-synonymous mutations in the *pmrA/pmrB* and *phoQ/phoP* operons and the *phoP/Q* regulator *mgrB* gene, leading to colistin resistance. The present data highlight the need for continuous monitoring of multidrug-resistant bacteria in wild animals to limit the spread of these threatening pathogens.

## 1. Introduction

Multidrug-resistant (MDR) pathogens, such as carbapenem-resistant Gram-negative bacteria (i.e., *Klebsiella pneumoniae*, *Acinetobacter baumannii*, *Salmonella enterica*, and *Enterobacter cloacae* complex bacterium), constitute a worldwide health threat [1]. Increased occurrence of infections caused by Gram-negative MDR bacteria, and a lack of new antibiotic drugs has led to the reevaluation of old antibiotics. As a result, colistin has become the last-line drug against serious bacterial infections, since it is effective against the majority of all multidrug-resistant Gram-negative bacteria [2]. However, a gradual increase in the prevalence of colistin resistance has been observed in various genera, including *Escherichia*, *Klebsiella*, *Salmonella*, *Shigella*, and *Enterobacter*, leading to a serious health threat [3].

Nowadays, two colistin resistance mechanisms are known in Gram-negative bacteria. One involves structural modifications of bacterial lipopolysaccharide, including the addition of 4-amino-4-deoxy-l-arabinose or phosphoethanolamine, following chromosomal mutations in genes encoding the two-component systems (phoP/Q and pmrA/B, or in the mgrB, a negative regulator of the PhoPQ system [4]). The other mechanism involves the phosphoethanolamine transferase *mcr* genes, a recently identified horizontally transferable plasmid-mediated colistin-resistance gene [5].

*Enterobacter cloacae* complex (ECC) is a member of the ‘ESKAPE’ group (*Enterococcus faecium*, *Staphylococcus aureus*, *Klebsiella pneumoniae*, *Acinetobacter baumannii*, *Pseudomonas aeruginosa*, and *Enterobacter* spp.) [1]. These pathogens are described as the leading cause of resistant nosocomial infections [1,6,7,8]. Among the *Enterobacter* genus, *Enterobacter cloacae* complex (ECC), including *E. cloacae*, *E. asburiae*, *E. hormaechei*, *E. kobei*, *E. ludwigii*, *E. mori*, *E. nimipressuralis*, *E. roggenkampii*, and *E. bugandensis*, are the most important clinically encountered pathogens with natural and/or acquired resistance to many antibiotics [2].

Previous studies demonstrated the presence of antimicrobial-resistance (AMR) genes in bacteria from a large variety of wildlife [9,10,11] and domestic [12,13] animals throughout Europe, including resistance to third generation cephalosporins, fluoroquinolones, carbapenems, and even colistin [9,10]. Due to intensive contact between humans and domestic and wild animals, the transmission of antimicrobial-resistant Gram-negative bacteria and/or interbacterial exchanges of AMR genes between bacteria from different niches is frequent [12,13]. In addition, some studies have highlighted the role of wild animals as bioindicators or sentinels for the propagation of resistant bacteria in the environment [14,15,16]. Consequently, the implementation of a “One Health” approach is timely for studying all underlying economic, social, political, environmental, and biological factors involved in the biology of bacteria carrying AMR genes, in order to identify key priorities for combating AMR pathogens [17,18,19]. Thus, the aim of this study was to investigate the origin of colistin resistance in *Enterobacter hormaechei* subsp. *steigerwaltii* isolated from fecal samples of French wild boar.

## 2. Results

### 2.1. Antimicrobial Susceptibility Testing

One bacterium from the *E. cloacae* complex (ECC) was isolated. The antimicrobial susceptibility test revealed that the isolated strain was susceptible to cefepime, piperacillin-tazobactam, ceftriaxone, ertapenem, imipenem, trimethoprim–sulfamethoxazol, ciprofloxacin, and gentamycin, while the strain was resistant to two antimicrobial classes represented by the β-lactams (i.e., amoxicillin, amoxicillin–clavulanic acid, and cephalothin) and polymyxins (i.e., colistin). The minimum inhibitory concentration of colistin was determined by broth microdilution, with an MIC = 4 mg/L.

### 2.2. Molecular Analysis

All PCR reactions yielded the amplification of the chromosomal genes of interest (i.e., *phoQ/phoP*, *pmrA/pmrB*, and *mgrB*) and housekeeping genes (i.e., *dnaA*, *fusA*, *gyrB*, *leuS*, *pyrG*, *rplB*, and *rpoB*). However, despite several attempts, none of the plasmid-mediated mobile colistin-resistance genes (*mcr*) were amplified.

The maximum likelihood phylogeny based on the chromosomal housekeeping genes (Figure 1) showed evidence that the isolated strain from wild boar is an integral part of *Enterobacter cloacae* complex bacteria and it clustered with the reference *E. hormaechei* subsp. *steigerwaltii* strain (GenBank accession number: CP017179, ST906). The MLST analysis performed on PubMLST server (https://pubmlst.org/ecloacae/, accessed on 15 July 2022) yielded the identification of this isolate as a new strain submitted under the accession ST1042.

By analogy to the colistin-sensitive-type strain of *E. asburiae* (*E. asburiae*, ATCC35953, GenBank accession number: CP011863), 5 non-synonymous mutations, in total, in the *phoP*, 17 in *phoQ*, 12 in *pmrA*, 24 in *pmrB*, and 2 in *mgrB* genes were recorded. Of those, two mutations (*phoP*: V5R and *pmrA*: D177E) were strain-specific and were considered as mutations affecting protein function according to the result of sorting intolerant from tolerant (SIFT), calculated on (https://sift.bii.a-star.edu.sg, accessed on 15 July 2022) (Figure 1 and Figure 2).

The parsimony tree performed on the mutation’s matrix showed that colistin resistance in ECC strains is related to the presence of intolerant mutations in the two-component systems (phoP/Q and pmrA/B and the mgrB genes), while the resistance profile (low, medium, high) is cluster dependent according to the MLST phylogeny (Figure 1).

## 3. Discussion

The emergence of new infectious pathogens of zoonotic concern in wildlife has increased general interest in wild animals [22]. However, studies on antimicrobial-resistant bacteria from wild fauna are scant as access to their biological samples is difficult. Here, we report, for the first time, a colistin-resistant strain of *E. hormaechei* subsp. *steigerwaltii* isolated from wild boar (*Sus scrofa*). Phenotypic and genotypic characterizations conducted in the current study emphasize the role of the inactivation in the two-component systems (phoP/Q and pmrA/B and the phoP/Q regulator mgrB gene) in the colistin-resistance mechanism from the *E. hormaechei* subsp. *steigerwaltii* strain.

Despite the large sample panel tested herein, only one antimicrobial-resistant ECC strain was isolated in wild boar from Southwest France. The low prevalence of resistant bacterial strains from wild boar was also reported in Germany [23], Spain, and Portugal [24], which may reflect both a low level of antimicrobial-resistant bacteria in these areas and the low exposure of these animals to antimicrobial drugs [25]. Nowadays, several studies using genomics have reported the occurrence of some *phoP/phoQ* and *pmrA/pmrB* profiles in ECC strains isolated from both humans and animals in several parts of the world (i.e., Japan, Netherlands, and USA), suggesting an anthropogenic origin for these pathogens. Moreover, wild animals are not treated directly with antimicrobial drugs, while the environmental exposure to antimicrobials could contribute to the selection of resistant bacteria in these animals, as reported in wild boar in Europe [5]. In addition, the expansion of urbanization to the detriment of forests has been reported to be another cause of contamination of wild fauna with antimicrobial-resistant bacteria through food, water, or direct contact with garbage and sewage [26], which may explain the carriage of a colistin-resistant ECC strain by wild boar in the present study.

Available studies have shown that the prevalence of bacteria and the results of the antimicrobial sensitivity analysis vary among wild species and their geographical locations [25]. Further analyses with respect to regional distribution and genetic traits as well as representative animal fauna need to be carried out to examine potential hosts and regional hot spots of AMR in wildlife in France.

Phenotypically, the *E. hormaechei* subsp. *steigerwaltii* herein isolated was β-lactams resistant (i.e., amoxicillin, amoxicillin–clavulanic acid, cephalothin). Β-lactam resistance is not new in *E. hormaechei* strains and has been proven to be linked to chromosomally encoded *AmpC* β-lactamases [27]. Genotypically, the chromosomal *phoP/phoQ* and *pmrA/pmrB* and *mgrB* genes revealed several non-synonymous mutations, particularly the V5R mutation in the *phoP* and D177E mutation in *pmrA* genes. Interestingly, these two mutations were strain specific and were involved in the alteration in protein function. Moreover, the absence of *mcr* genes from the isolated strain suggests that two-component systems (phoP/Q and pmrA/B and mgrB) are responsible for the observed colistin resistance.

Despite the important number of studies describing colistin-resistance mechanisms in ECC bacteria, there is limited information on the patterns mediating these mechanisms in ECC bacteria [3,12]. Mushtaq and colleagues investigated relationships to species, genome, carbon source utilization, and LPS structure on 1749 ECC strains [3]. Authors reported that colistin resistance is associated with particular genomic and metabolic clusters inducing changes in LPS architectures, which is directly linked to the chromosomal mutations in genes encoding the two-component systems (phoP/Q and pmrA/B or in the phoP/Q regulator mgrB gene) [4]. However, genomic data from this study were not available [3]. On the other hand, and despite the carriage of different *mcr* variants by the ECC bacteria, only the *mcr-10* variants were statistically linked to colistin resistance or reduced susceptibility to colistin [28,29], without a clear confirmation of this statement [30]. The present study highlighted that the colistin-resistance profile in ECC strains is dependent on phylogenetic clusters and to mutations affecting protein function of the two two-component systems (phoP/Q and pmrA/B or in the phoP/Q regulator mgrB gene) as previously reported by using genome-based phylogeny [3]. However, the colistin-resistance mechanism remains unexplained genomically in some species. For example, the S/R-colistin strain (GenBank accession: CP010512) showed a completely independent colistin response regarding the genomic context [31,32,33].

## 4. Materials and Methods

### 4.1. Sample Processing and Antimicrobial Susceptibility Testing

In 2016, 358 fecal samples of wild boar (*Sus scrofa*) were collected in the military camp of Canjuers (43°42′17.99′′ N 6°18′18.00′′ E) in the Var (Southeast France). The selective Lucie Bardet-Jean-Marc Rolain (LBJMR) medium (S177) was used for the isolation and culture of ECC isolates as described elsewhere [34]. Broth microdilution for antimicrobial susceptibility testing was performed according to the European Committee on Antimicrobial Susceptibility testing breakpoints.

### 4.2. DNA Extraction and Sequencing

The isolated bacterium was subjected to DNA extraction using the Biorobot EZ1 System with the EZ1 DNA tissue kit (Qiagen, Courtaboeuf, France) following the manufacturer’s recommendations. Genomic DNA was subjected to PCR amplification and sequencing targeting three group of genes: (i) housekeeping genes (i.e., *dnaA*, *fusA*, *gyrB*, *leuS*, *pyrG*, *rplB*, and *rpoB*) for multiloci sequence typing; (ii) the chromosomal *phoP/phoQ* and *pmrA/pmrB* and *mgrB* genes; and (iii) a group of known plasmid-mediated colistin-resistance (mcr) genes. PCR amplification was confirmed in a 2% agarose gel with ethidium bromide. The PCR products of all positive reactions were purified by filtration using NucleoFast 96 PCR DNA purification plate prior to the BigDye reaction using the Terminator v3.1 Cycle Sequencing Kit (Applied Biosystems, Foster City, CA, USA). The BigDye products were purified on the Sephadex G-50 Superfine gel filtration resin prior to sequencing on the ABI Prism 3130XL.

### 4.3. Molecular Analysis

#### 4.3.1. Data Preparation

To characterize the isolated bacterium, a dataset of 31 ECC strains was selected on the basis of the availability of information on MIC of colistin and genomic and plasmidic sequences. Of those, 12 bacterial genomes representing 5 ECC species (i.e., *E. hormaechei*, *E. roggenkampii*, *E. cloacae*, *E. kobei*, and *E. asburiae*) were available only in paired reads from GenBank Databse (Project accession: PRJDB13693) [3]. Genome assemblies were generated using a pipeline grouping different software (i.e., Velvet [35], Soap Denovo [36], and Spades [37]) as described elsewhere [38]. To strengthen the molecular phylogeny, 10 types of ECC strains were also involved in the study (Table 1). Briefly, the housekeeping genes of interest (i.e., *dnaA*, *fusA*, *gyrB*, *leuS*, *pyrG*, *rplB*, and *rpoB*) were researched and retrieved from the selected genomes and were then blasted on PubMLST server (https://pubmlst.org/ecloacae/, accessed on 15 July 2022) to confirm and/or identify the sequence type, while protein-coding genes in the two-component systems (pmrA/B and phoQ/P and the phoP/Q regulator mgrB gene) were also searched and retrieved from the selected strains.

#### 4.3.2. Molecular Characterization

Sequence alignment was performed using MAFFT [39]. The Bioedit software was used to manually refine the multisequence alignments [40] prior to sequence concatenation using SEAVIEW [41]. The multisequence alignment was then subjected to maximum-likelihood-based phylogeny using iqtree2 software [42]. The best-fit model was selected using model finder [43] to compute the tree under 1000 bootstrap replications. Sequence of *Klebsiella aerogenes*, strain KCTC2190 (GenBank accession: CP002824) was used as outgroup to root the tree.

In addition, the parsimony tree was performed on the mutation matrix of the two-component systems (pmrA/B and phoQ/P and the phoP/Q regulator mgrB genes). Briefly, protein sequences were retrieved from all genomes of bacterial strains for which the results of colistin testing were available (n = 31) and were compared to sequences from the colistin-sensitive reference *E. asburiae* (GenBank accession: CP011863) using ClusterW and PROVEAN [44]. Identified mutations were then subjected to sorting intolerant from tolerant (SIFT) calculated on (https://sift.bii.a-star.edu.sg, accessed on 15 July 2022). The SIFT score and mutation matrix were then subjected to parsimony tree using PARS and CONSENSE applications within PHYLIP program [21]. The resulting heatmap as well as the information on colistin sensitivity of each strain (i.e., number of mutations per gene in the two-component systems (pmrA/B and phoQ/P and the phoP/Q regulator mgrB), MIC for colistin, and the presence of *mcr* genes) were used to annotate the tree using iTOL software [20].

Finally, the SMART server [45] was used to predict protein domains in the two-component systems (pmrA/B and phoQ/P and the phoP/Q regulator mgrB) using *Escherichia coli* K-12 sub-strain MG1655 as type strain.

**Table 1 pathogens-11-01022-t001:** Description of the ECC strains used in the molecular analysis.

Strain Identification	Acc. No.	Strain	ST. Acc.	*dnaA*	*fusA*	*gyrB*	*leuS*	*pyrG*	*rplB*	*rpoB*	Source	MIC (mg/L)	*Mcr* Genes	References
*E. h.* subsp. *steigerwaltii*	ST1042	B-107	1042	4	6	4	77	11	36	39	wild boar	4	No	This study
* E. h. * subsp. *steigerwaltii* ^a^	CP017179	DSM 16691	906	58	174	4	6	42	4	25				[46]
*E. h. subsp. steigerwaltii*	CP083849	14269	125	63	3	66	68	3	16	3		8	mcr-9.2	[30]
* E. h. * subsp. *hoffmannii* ^a^	CP017186	DSM 14563	816	59	9	80	172	35	6	6				[46]
* E. h. * subsp. *oharae* ^a^	CP017180	DSM 16687	108	68	8	75	63	65	34	35				
*E. hormachei*	DRX366480	En42	1579	4	4	4	6	72	4	6	Dog	>128	No	[47]
*E. roggenkampii*	CP083853	12795	523	36	39	192	206	49	12	20		>128	mcr-9.2	[30]
*E. roggenkampii*	CP083819	13840	702	36	25	49	30	49	21	143		16	mcr-10.1	
*E. roggenkampii*	DRX366478	En37	1576	72	278	71	383	160	46	172	Dog	>128	mcr-10	[47]
*E. roggenkampii*	DRX366479	En50	606	37	27	49	57	200	21	20	Cat	>128	No	
* E. oligotrophica * ^a^	AP019007	CCA6	Novel	401 ^b^	266	423	496 ^b^	342 ^b^	4	273	leaf soil			[46]
* E. roggenkampii * ^a^	CP017184	DSM 16690	Novel	270	39	91	92	312	12	26				
*E. xiangfangenisc* ^a^	CP017183	LMG 27195	544	10	21	9	44	45	4	33				
*E. c.* subsp. *cloacae* ^a^	CP0011918	ATCC 13047	873	85	63	101	103	96	6	53	Human	8	No	
*E. c. subsp. cloacae*	CP083821	12961	84	60	1	61	1	36	22	1		>128	mcr-10.1	[30]
*E. cloacae*	DRX366481	En46	765	156	92	169	218	105	22	99	Dog	>128	No	[47]
*E. cloacae*	CP010512	colR/S	252	22	15	102	104	101	11	10	Human	1/500	No	[31]
*E. cloacae*	CP032291	/0073	73	8	33	6	9	12	6	8	Human	>8	No	[48]
*E. cloacae*	CP021749	163	163	71	3	87	89	13	16	3	Human	>8	No	
*E. cloacae*	CP014280	MBRL1077	Novel	467 ^b^	202	484 ^b^	582 ^b^	377	219	266	Human	> 4	No	[49]
* E. kobei * ^a^	CP017181	ATCC BAA-260	806	71	3	87	312	254	16	167				[46]
*E. kobei*	CP083828	11778	280	3	3	58	37	3	16	17		>128	mcr-10.2	[30]
*E. kobei*	DRX366470	En3	591	3	3	110	232	19	16	17	Dog	>128	No	[47]
*E. kobei*	DRX366471	En4	591	3	3	110	232	19	16	17	Dog	>128	No	
*E. kobei*	DRX366472	En5	591	3	3	110	232	19	16	17	Dog	>128	No	
*E. kobei*	DRX366473	En14	591	3	3	110	232	19	16	17	Cat	>128	No	
*E. kobei*	DRX366474	En49	1577	316	277	110	518	3	16	210	Dog	>128	No	
*E. kobei*	CP083862	11743	56	42	3	52	37	23	16	3		>128	mcr9.1/2 copies	[30]
*E. kobei*	CP083857	12379	57	43	3	51	36	18	16	19		>128	mcr-9.2	
*E. asburiae* ^a^	CP011863	ATCC 35953	807	255	166	280	313	255	11	166	Human	1	No	[3]
*E. asburiae*	DRX366475	En6	1578	229	14	235	519	98	11	16	Dog	>128	No	[47]
*E. asburiae*	DRX366476	En19	562	22	15	102	104	101	11	71	Cat	>128	No	
*E. asburiae*	DRX366477	En30	1578	229	14	235	519	98	11	16	Cat	>128	mcr-9	
*E. asburiae*	CP083842	16773	41	37	25	49	30	49	21	20		>128	mcr-9.1	[30]
*E. asburiae*	CP083834	AR0468	27	26	16	25	53	22	9	15		>128	mcr-9.1	
*E. asburiae*	CP083830	AR2284	252	22	15	102	104	101	11	10		>128	mcr-9.1 + mcr-9.2	
*E.asburiae*	AP022628	A2563	484	26	14	143	191	61	11	89	Human	0.125	*mcr-9*	[50]
*E. asburiae*	CP083815	161373	1	1	1	1	1	1	1	1		16	mcr-10.1	[30]
* E. ludwigii * ^a^	CP017279	EN-119	714	13	2	105	133	51	2	14				[46]
*E. ludwigii*	CP083824	11894	Novel	280 ^b^	15	318 ^b^	361 ^b^	293	106	156		128	mcr-10.4	[30]

ST: sequence type; MIC: minimum inhibitory concentration; ^a:^ indicates type ECC strains; ^b:^ indicates the most closest allele found in PubMLST database.

## 5. Conclusions

Our results demonstrate that wild boar could be colonized by colistin-resistant *E. hormaechei* subsp. *steigerwaltii*, highlighting their potential role as reservoirs of AMR bacteria. Because of the consumption of wild boar as game animal as well as their proximity to domestic animals and farms, these common animals could be a zoonotic source for transmission of colistin-resistant bacteria to humans.

## Figures and Tables

**Figure 1 pathogens-11-01022-f001:**
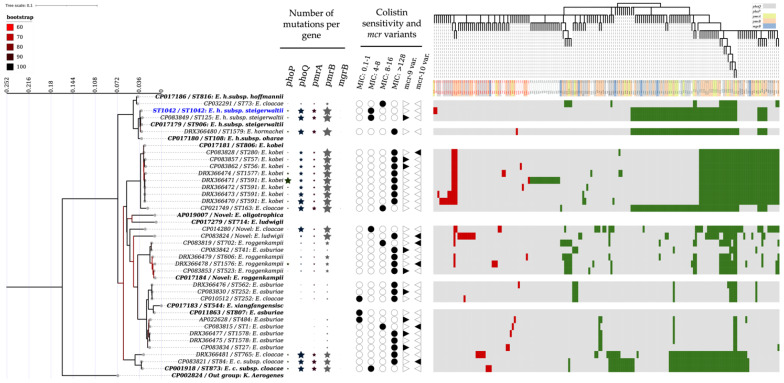
Molecular characterization of the *E. hormaechei* subsp. *steigerwaltii* strain ST1042 isolated from wild boar in the present study. Maximum likelihood (ML) phylogeny showing the position of ST1042 strain among the member of the ECC strains. The tree corresponds to the IQTREE inferred from the 41 concatenated (3509 bps) sequences (i.e., *dnaA*, *fusA*, *gyrB*, *leuS*, *pyrG*, *rplB*, and *rpoB*) with 17.6% of informative sites. Branches are color coded according to the bootstrap values. The tree was rooted at the midpoint using iTOL v5 software [20]. Accession numbers, species names, and ST accessions are indicated at the tip of each branch. The bold blue label indicates the sequence obtained in this study. Bold black labels indicate reference strains. Size-dependent stars indicate the number of mutations identified in the two-component system genes (phoP/Q, pmrA/B, and mgrB) comparatively to the reference colistin-sensitive strain *E. asburiae* (CP011863). Black-filled circles indicate the colistin profile according to the MIC expressed in mg/L. The presence of *mcr* gene is indicated by the filled black right pointing (mcr-9 variants) and the left pointing (mcr-10 variants) triangles. The heatmap shows the profile of each strain according to the identified mutation in the two-component system genes (phoP/Q, pmrA/B, and mgrB). Green areas represent tolerated mutation, while red areas represent intolerant mutations according to the result of sorting intolerant from tolerant (SIFT) calculated on (https://sift.bii.a-star.edu.sg, accessed on 15 July 2022). The horizontal dendrogram corresponds to the parsimony tree generated by the PARS and CONSENSE applications within the PHYLIP program [21]. Red-colored labels and the color-coded background indicate intolerant mutation names and their origine (i.e., *phoP/Q*, *pmrA/B*, and *mgrB*), respectively.

**Figure 2 pathogens-11-01022-f002:**
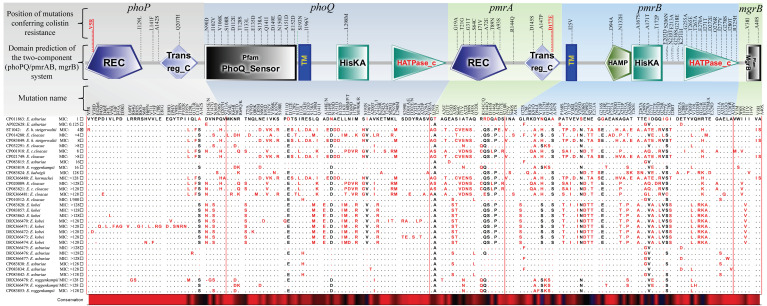
Location of the non-synonymous mutations throughout the predicted domains of the two-component systems (phoP/Q and pmrA/B and the phoP/Q regulator mgrB gene). Black and red texts indicate, respectively, tolerated mutations and mutations affecting protein function according to the result of sorting intolerant from tolerant (SIFT) calculated on (https://sift.bii.a-star.edu.sg, accessed on 15 July 2022). The CLUSTALW alignment represents the informative sites of protein sequences of the *phoP/phoQ* and *pmrA/pmrB* and the *PhoPQ* regulator *mgrB* genes of ECC strains.

## Data Availability

The housekeeping (i.e., *dnaA*
*fusA*, *gyrB*, *leuS*, *pyrG*, *rplB*, and *rpoB*) genes amplified in the present study are available from PubMLST database under the accession number: ST1042. Descriptive pipeline and dataset used in the present study are available as a GitHub repository: https://github.com/YLdz-SM/CLT-E.h.steigerwaltii.git (accessed on 15 July 2022).

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
