# Peer review of "Colistin Resistance Mechanism in Enterobacter hormaechei subsp. steigerwaltii Isolated from Wild Boar (Sus scrofa) in France"

_pathogens, 2022, doi:10.3390/pathogens11091022_

Round 1

Reviewer 1 Report

Authors sought for Colistin resistant Enterobacter in 358 fecal samples of wild boar and found only one positive strain which was identified as  E. hormaechei subsp. steigerwaltii. By a molecular analysis, they could find several mutations likely affecting the activity of proteins phoP/phoQ and pmrA/pmrB, and mgrB. Moreover they could identified new mutations (V5R mutation in phoP and 143 D177E in pmrA) belonging to this category which could participate to the colistin resistance. The experiments are well described and the article is clearly written. It shows that some colistin resistant enterobacteria coming from humans could colonise wild animals.

There is no commentaries about the MIC of this isolated strain ST1042 : 4µg/ml is quite low. To my opinion it requires some comparisons to most of colistin resistant Enterobacter whose MIC are rather > 128µg/ml.

Minor points :

L203  Database

L 89 repetition of  « to the »

L113 space is lacking after [23]

L92 revealed

L162 Genomically

L163 Genbank

L172 remove « 2017 » ?

Author Response

Comments and Suggestions for Authors

Authors sought for Colistin resistant Enterobacter in 358 fecal samples of wild boar and found only one positive strain which was identified as E. hormaechei subsp. steigerwaltii. By a molecular analysis, they could find several mutations likely affecting the activity of proteins phoP/phoQ and pmrA/pmrB, and mgrB. Moreover they could identified new mutations (V5R mutation in phoP and 143 D177E in pmrA) belonging to this category which could participate to the colistin resistance. The experiments are well described and the article is clearly written. It shows that some colistin resistant enterobacteria coming from humans could colonise wild animals.

We grateful for the reviewer for his positive feedback on our manuscript. We addressed bellow, a point-by-point response to all comments (please see the tracked version of the new MS).

There is no commentaries about the MIC of this isolated strain ST1042 : 4µg/ml is quite low. To my opinion it requires some comparisons to most of colistin resistant Enterobacter whose MIC are rather > 128µg/ml.

Response: Indeed, the MIC of 4 mg/L is low, however, it represent a resistant colistin profile as the breakpoint is at 2 mg/L. Concerning the comparison with other strains, we really would like to made this comparison, however genomic data are lacking for most strains reported as colistin-resistant, For example, no available genomes among the 51 colistin resistant strains despite the indication of the project accession number in the study of Mushtaq et al., (Sequence data supporting these analyses are in the process of being made available in the European Nucleotide Archive, under project ac- cession number PRJEB35697) which remains the most exhaustive study in term of strain diversity.

Mushtaq, S.; Reynolds, R.; Gilmore, M.C.; Esho, O.; Adkin, R.; García-Romero, I.; Chaudhry, A.; Horner, C.; Bartholomew, T.L.; Valvano, M.A. Inherent Colistin Resistance in Genogroups of the En-terobacter Cloacae Complex: Epidemiological, Genetic and Biochemical Analysis from the BSAC Re-sistance Surveillance Programme. Journal of Antimicrobial Chemotherapy 2020, 75, 2452–2461.

Minor points

L203  Database

Response: Done.

L 89 repetition of  « to the »

Response: Done.

L113 space is lacking after [23]

Response: Done.

L92 revealed

Response: Done.

L162 Genomically

Response: Done.

L163 Genbank

Response: Done.

L172 remove « 2017 » ?

Response: Done.

Reviewer 2 Report

Manuscript ID: pathogens-1894787

General and Specific comments:

The study is original, contributing to the existing knowledge regarding antibiotic resistant bacteria and genes in wild animals, and their impact at the environment-livestock-human interface.

Some points that deserve revision are detailed below:

1. The authors refer a “…multidrug-resistant (amoxicillin, amoxicillin-clavulanic acid, cephalothin and colistin) Enterobacter hormaechei subsp. steigerwaltii…”. However, only two antimicrobial classes are represented in this resistance phenotype (β-lactams and polymyxins) and it is known that Enterobacter hormaechei subsp. steigerwaltii has intrinsic resistance to some β-lactams due to chromosomally encoded AmpC β-lactamases (https://doi.org/10.1016/j.ijantimicag.2022.106650). Could we really classify it as a multidrug-resistant (MDR) strain, according to standard definition of MDR bacteria? (DOI: 10.1111/j.1469-0691.2011.03570.x).

INTRODUCTION

2. Lines 26, 29, 30, 36, and 56: Replace “gram” by “Gram”.

3. Lines 26-28: A more recent and embracing reference should be provided.

4. Lines 29-30: Please avoid the repeating of “Gram-negative pathogens” on the statement.

5. Line 56: The authors refer “the transmission of resistant bacteria”, but it could be the transmission of antimicrobial resistant bacteria and/or genes.

RESULTS

6. Lines 72-73: The method used for susceptibility testing should be referred in the Material and Methods section. “concentration” instead of “concentrations”. And “of 4 mg/L” instead of “= 4 µg/mL”.

7. Line 85: Replace “type species” by “type strain”.

8. Line 86: The term “of” is in duplicate.

9. Line 87: Replace “two” by “2”, and please finish the sentence.

10. Line 89: The terms “to the” are in duplicate.

11. Lines 91-94: Please improve the statement, to clarify the message.

MATERIAL AND METHODS

12. Line 181: Acronyms should be preceded by the full term when used for the first time. A reference should also be indicated for this culture media (the authors state “as described elsewhere”, but did not give a reference). Replace “ECC bacteria” by “ECC isolates”.

13. Lines 182-183: Please provide the method used (disk diffusion, broth microdilution, other?). Also, a reference should be provided. 

14. Lines 185 and 199: Replace “bacterium” by “bacteria” (verify in other sections of the paper).

15. Lines 188-189: The housekeeping gene “dnaA” was not considered for the MLST scheme?

16. Line 190: “Mobile” could be removed, as it is inferred by the expression “plasmid-mediated”.

17. Line 199: Are they 31 or 30 ECC strains (excluding the ECC strain isolated during this study)?

18. Lines 205-206: There are 7 or 10 ECC type strains? In Table 1, 10 ECC strains are indicated as ECC type strains. Please verify.

19. Line 212: Subsection number is 4.3.2.

20. Line 225: “mutations” instead of “mutation”.

21. Line 230: “for colistin”.

DISCUSSION

22. The fact that this study constitutes the first report of the isolation of a colistin-resistant E. cloacae complex in wild boars should be clarified earlier in the discussion.

23. Lines 120-123: This long statement needs some punctuation marks.

24. Line 124: Enterobacterial or ECC?

25. Line 125: “This corroborates…”. The author should specify what indicates as “This”.

26. Lines 126-128: This statement is a transcription from Reference 25.

27. Lines 128-131: Please improve the English.

28. Lines 133-138: The statement is too long and difficult to follow. Please clarify.

29. Line 139: In “the present” study?

30. Lines 144-145: “…which characterizing the isolated strains and showing effect on…”. Please improve the English.

31. If authors are referring to the two compound systems, and not to the corresponding genes, their designation do not need to be italicized (protein designation nomenclature).

32. Lines 148-150: Please provide reference(s).

CONCLUSIONS

33. Other possible transmission routes to humans should be considered.

TABLE

34. In the title, replace “strain” by “strains”. Please revise the nomenclature for designation of bacteria that appears in the column “Precise species names”. This column also deserves a more appropriate title, as there are designations beyond species (subspecies) (for example, it could be replaced by “Strain identification” or “Bacterial identification”).

35. In the footnote, replace “closet” by “closest”. Acronyms such as ST and MIC should appear in full term in the footnote.

36. Replace “mcg/ml” by “mg/L”.

37. Is one of the MIC values 1/500?

FIGURES

38. Figure 1: Line 104 – Please replace “µg/ml” by “mg/L”; Line 105 – Please replace “showing” by “shows”; Line 109 - Please replace “corresponds” by “correspond”; Lines 109-110 – this sentence contains information provided in previous statement.

39. Figure 2: Line 113 – Reference 23 should be repositioned.

Author Response

Comments and Suggestions for Authors

Manuscript ID: pathogens-1894787

General and Specific comments:

The study is original, contributing to the existing knowledge regarding antibiotic resistant bacteria and genes in wild animals, and their impact at the environment-livestock-human interface.

We thank the reviewer for his valuable feedback on our manuscript. Here is a point-by-point response to the comments and concerns (please see the tracked version of the new MS).

Some points that deserve revision are detailed below:

  1. The authors refer a “…multidrug-resistant (amoxicillin, amoxicillin-clavulanic acid, cephalothin and colistin) Enterobacter hormaechei subsp. steigerwaltii…”. However, only two antimicrobial classes are represented in this resistance phenotype (β-lactams and polymyxins) and it is known that Enterobacter hormaechei subsp. steigerwaltiihas intrinsic resistance to some β-lactams due to chromosomally encoded AmpC β-lactamases (https://doi.org/10.1016/j.ijantimicag.2022.106650). Could we really classify it as a multidrug-resistant (MDR) strain, according to standard definition of MDR bacteria? (DOI: 10.1111/j.1469-0691.2011.03570.x).

Response: We thank the reviewer for pointing out this issue. Indeed, the MDR bacteria referred to those resistant to at least three antimicrobial classes, which is not the case of the Enterobacter hormaechei subsp. steigerwaltii herein isolated. We have now corrected the terminology to be “antimicrobial-resistant strain instead of MDR strain” (Line 16-17, and elsewhere in the MS).

INTRODUCTION

  1. Lines 26, 29, 30, 36, and 56: Replace “gram” by “Gram”.

Response: Done

  1. Lines 26-28: A more recent and embracing reference should be provided.

Response: Done

  1. Lines 29-30: Please avoid the repeating of “Gram-negative pathogens” on the statement. 

Response: We have made this correction.

  1. Line 56: The authors refer “the transmission of resistant bacteria”, but it could be the transmission of antimicrobial resistant bacteria and/or genes.

 Response: We have completed by this information as requested (Line 57-59, new MS).

RESULTS

  1. Lines 72-73: The method used for susceptibility testing should be referred in the Material and Methods section.

Response: We have moved the text to the Material and Methods section.

“concentration” instead of “concentrations”. And “of 4 mg/L” instead of “= 4 µg/mL”.

Response: Done.

  1. Line 85: Replace “type species” by “type strain”.

Response: Done.

  1. Line 86: The term “of” is in duplicate.

Response: Done.

  1. Line 87: Replace “two” by “2”, and please finish the sentence.

Response: We thank the reviewer for pointing out our inconsistency in writing.

  1. Line 89: The terms “to the” are in duplicate.

Response: Done.

  1. Lines 91-94: Please improve the statement, to clarify the message.

Response: Thank you for pointing out this issue. We have now altered the text to be: “The parsimony tree performed on the mutation’s matrix showed that colistin resistance of ECC strains is related to the presence of intolerant mutations in the two compound systems (phoP/Q and pmrA/B and the mgrB genes), while the resistance profile (low, medium, high) is cluster dependent according to the MLST phylogeny (Fig. 1)” (Line 96-99, new MS)..

MATERIAL AND METHODS

  1. Line 181: Acronyms should be preceded by the full term when used for the first time. A reference should also be indicated for this culture media (the authors state “as described elsewhere”, but did not give a reference). Replace “ECC bacteria” by “ECC isolates”. 

Response: All these informations are now added and the text was altered accordingly (Line 216-19, new MS).

  1. Lines 182-183: Please provide the method used (disk diffusion, broth microdilution, other?). Also, a reference should be provided.  

Response: We have now provided these details (Line 219-21, new MS).

  1. Lines 185 and 199: Replace “bacterium” by “bacteria” (verify in other sections of the paper).

Response: Here, we are referring to our isolated strain. We believe that the term bacterium is more appropriate as we isolated only one strain.

  1. Lines 188-189: The housekeeping gene “dnaA” was not considered for the MLST scheme? 

Response: Thank you for pointing out this missing. Indeed, the dnaA gene was also involved in the MLST, and we have now mention it in the text.

  1. Line 190: “Mobile” could be removed, as it is inferred by the expression “plasmid-mediated”.

Response: Done.

  1. Line 199: Are they 31 or 30 ECC strains (excluding the ECC strain isolated during this study)? 

Response: A total of 31 strains with known MIC for colistin were used along the strain herein we isolated.

  1. Lines 205-206: There are 7 or 10 ECC type strains? In Table 1, 10 ECC strains are indicated as ECC type strains. Please verify.

Response: We thank the reviewer for pointing out this error. We have now clarified that 10 reference strains were used (Line 244, new MS).

  1. Line 212: Subsection number is 4.3.2.

Response: Done.

  1. Line 225: “mutations” instead of “mutation”.

Response: Done.

  1. Line 230: “forcolistin”.

 Response: Done.

DISCUSSION

  1. The fact that this study constitutes the first report of the isolation of a colistin-resistant E. cloacae complex in wild boars should be clarified earlier in the discussion.

Response: Done (Line 131-2, new MS).

  1. Lines 120-123: This long statement needs some punctuation marks.

Response: Done (Line 133-6, new MS).

  1. Line 124: Enterobacterial or ECC?

Response: Done.

  1. Line 125: “This corroborates…”. The author should specify what indicates as “This”.

Response: Done (Line 143-5, new MS).

  1. Lines 126-128: This statement is a transcription from Reference 25.

Response: We have adapted the text as requested (Line 145-7, new MS).

  1. Lines 128-131: Please improve the English.

Response: Done (Line 149-154, new MS).

  1. Lines 133-138: The statement is too long and difficult to follow. Please clarify.

Response: Done (Line 154-163, new MS).

  1. Line 139: In “the present” study?

Response: Done.

  1. Lines 144-145: “…which characterizing the isolated strains and showing effect on…”. Please improve the English.

Response: Done (Line 176-8, new MS)...

  1. If authors are referring to the two compound systems, and not to the corresponding genes, their designation do not need to be italicized (protein designation nomenclature).

Response: Done.

  1. Lines 148-150: Please provide reference(s).

 Response: Done.

CONCLUSIONS

  1. Other possible transmission routes to humans should be considered.

  Response: We have now provided the other possible transmission routes.

TABLE

  1. In the title, replace “strain” by “strains”. Please revise the nomenclature for designation of bacteria that appears in the column “Precise species names”. This column also deserves a more appropriate title, as there are designations beyond species (subspecies) (for example, it could be replaced by “Strain identification” or “Bacterial identification”).

 Response: Done.

  1. In the footnote, replace “closet” by “closest”. Acronyms such as ST and MIC should appear in full term in the footnote.

Response: Done.

  1. Replace “mcg/ml” by “mg/L”.

Response: Done.

  1. Is one of the MIC values 1/500?

 Response: Yes, this strain (from Band et al.,) is considered as colistin-sensitive (MIC= 1 mg/L) and colistin-resistant (MIC= 500 mg/L) at the same time according to the phenotypic profile, while both strains share the same genome.

Band, V.I.; Crispell, E.K.; Napier, B.A.; Herrera, C.M.; Tharp, G.K.; Vavikolanu, K.; Pohl, J.; Read, T.D.; Bosinger, S.E.; Trent, M.S.; et al. Antibiotic Failure Mediated by a Resistant Subpopulation in Enterobacter Cloacae. Nat Microbiol 2016, 1, doi:10.1038/nmicrobiol.2016.53.

FIGURES

  1. Figure 1:Line 104 – Please replace “µg/ml” by “mg/L”; Line 105 – Please replace “showing” by “shows”; Line 109 - Please replace “corresponds” by “correspond”; Lines 109-110 – this sentence contains information provided in previous statement.

Response: We have now addressed all the requested modifications.

  1. Figure 2:Line 113 – Reference 23 should be repositioned.

Response: Done.
